# Evolution of Cu-In Catalyst Nanoparticles under Hydrogen Plasma Treatment and Silicon Nanowire Growth Conditions

**DOI:** 10.3390/nano13142061

**Published:** 2023-07-12

**Authors:** Weixi Wang, Éric Ngo, Pavel Bulkin, Zhengyu Zhang, Martin Foldyna, Pere Roca i Cabarrocas, Erik V. Johnson, Jean-Luc Maurice

**Affiliations:** 1Laboratoire de Physique des Interfaces et Couches Minces, École Polytechnique, CNRS, IPParis, 91120 Palaiseau, Francemartin.foldyna@polytechnique.edu (M.F.);; 2Laboratoire LuMIn, École Normale Supérieure Paris-Saclay, CentraleSupélec, Université Paris-Saclay, CNRS, 91190 Gif-sur-Yvette, France

**Keywords:** Cu-In nanoparticle, plasma treatment, silicon nanowire, PECVD, TEM

## Abstract

We report silicon nanowire (SiNW) growth with a novel Cu-In bimetallic catalyst using a plasma-enhanced chemical vapor deposition (PECVD) method. We study the structure of the catalyst nanoparticles (NPs) throughout a two-step process that includes a hydrogen plasma pre-treatment at 200 °C and the SiNW growth itself in a hydrogen-silane plasma at 420 °C. We show that the H_2_-plasma induces a coalescence of the Cu-rich cores of as-deposited thermally evaporated NPs that does not occur when the same annealing is applied without plasma. The SiNW growth process at 420 °C induces a phase transformation of the catalyst cores to Cu_7_In_3_; while a hydrogen plasma treatment at 420 °C without silane can lead to the formation of the Cu_11_In_9_ phase. In situ transmission electron microscopy experiments show that the SiNWs synthesis with Cu-In bimetallic catalyst NPs follows an essentially vapor-solid–solid process. By adjusting the catalyst composition, we manage to obtain small-diameter SiNWs—below 10 nm—among which we observe the metastable hexagonal diamond phase of Si, which is predicted to have a direct bandgap.

## 1. Introduction

Semiconductor nanowires (NWs) are attracting increasing attention because of their application prospects in electronic, sensing, optoelectronic, and photovoltaic devices [1,2,3,4,5,6,7]. The most common methods for preparing these nano-objects are based on the vapor–liquid–solid (VLS) and vapor–solid–solid (VSS) growth processes, where the NW precipitates form a catalyst particle whose constituting elements are maintained in supersaturation [8]. The electrical and optical properties of the NWs are strongly size- and structure-dependent [9,10,11,12], and catalyst NPs play a vital role in determining the NW size, structure, morphology, distribution, and orientation. In this respect, using compound catalysts gives an additional degree of freedom to monitor these parameters: for example, Ga-Au [13] and Cu-Au [14] catalysts have been proven advantageous in adjusting SiNW geometrical shape, size, and orientation; Bi-Sn [15] catalysts can lead to a doping effect in SiNWs; Al-Au [16] catalysts yield reliable epitaxial growth of SiNWs on single-crystalline silicon substrates; Ag-Au [17] catalysts are employed to form compositionally abrupt interfaces of Si/Ge/Si heterojunctions in NWs. In our previous works on the mechanisms of growth of SiNWs [18,19], we have illustrated how the use of a mixed catalyst—where one of the components is liquid and the other one is solid—increases even more these possibilities, in a mode we have called liquid-assisted-VSS (LAVSS). Since then, the tuning of the shape or the orientation of the nanowires by LAVSS has also been used in the growth of compound semiconductor NWs [20]. The system we explored in the refs. [18,19] was that of Cu-Sn, which resulted in catalysts made of solid Cu_3_Si and liquid Sn during growth. With this type of catalyst, we were able to prepare SiNWs with the metastable hexagonal 2H structure [19] that is predicted to have a direct bandgap [21]. Indium, which has a similar phase diagram with Cu [22], possesses an even lower melting point than Sn (157 °C vs. 232 °C) and appealed to us as the next candidate to be associated with Cu. If, as a member of column 3 in the periodic table, it is a p-type dopant for Si, its solubility in Si is very low [23], so that its benefit as a catalyst overpasses, at least at this stage of our study, the drawback of a possible unwanted doping. Let us mention that the control of its introduction into the Si lattice during growth, through a carefully designed catalyst composition, would allow one an additional control of the NW properties [24].

Copper-indium alloys have been studied a lot, as they are used in catalysis, thin-film solar cells manufacturing, and interconnections in the electronics industry [25,26,27,28,29,30,31]. The reactions of Cu-In alloys as catalysts for electrochemical reduction of CO_2_ [32,33,34], as precursors for the growth of CuInSe_2_ solar cells [35,36,37], and as solders for the fabrication of Cu/In/Cu joints for silicon interconnection components [38,39,40], have been widely studied. Recently, it has been reported that Cu-In bimetallic NPs have been successfully used as a catalyst for the growth of SiNWs by a plasma-enhanced CVD (PECVD) method [41]. Therefore, a detailed characterization of Cu-In catalyst NPs is required for a better understanding of their influence on the SiNW growth mechanism. In addition, since the impact of plasma on the catalyst NPs [42] brings an additional complexity to the reactions between Cu and In for the synthesis of SiNWs at relatively low temperature, exploring the structure of the Cu-In NPs under various plasma conditions during NW growth is of great importance for a better control of SiNW fabrication. In this work, we study SiNW growth by using a bimetallic catalyst of Cu-In. To understand the growth mechanism, we observe the catalyst structure evolution by ex situ TEM throughout the SiNW growth process in a PECVD reactor; we also use in situ TEM to visualize the critical stage of plasma ignition and analyze the catalyst structure during SiNW growth. We demonstrate the phenomenon of crystallization and phase transformation in Cu-In NPs induced by hydrogen and hydrogen-silane plasmas. On the other hand, Cu-In alloys are widely used nano-solder materials as they can form small-diameter NPs [40,43], which is promising to obtain small-diameter SiNWs with the metastable 2H hexagonal diamond phase [19,44]. We report in this study the discovery of the 2H phase in the SiNWs synthesized with the Cu-In catalyst.

## 2. Materials and Methods

The bimetallic Cu-In NPs were prepared in a BOC Edwards Auto 306 Evaporator FL 400. Cu was first evaporated on the substrates; then, the Cu-loaded crucible was changed for an In-loaded one and In was evaporated on top of Cu. The nominal thickness of each of the evaporated elements was thus measured independently by a quartz microbalance in the thermal evaporator. The sensitivity of the quartz microbalance, together with the speed at which the experimentalist moves the shutter when the desired value is displayed, lead to a limited precision in the deposited matter that may significantly vary from one deposit to the next. Based on energy-dispersive X-ray spectroscopy performed on deposits made on carbon-coated gold TEM grids, we evaluated possible departures from the nominal deposited value to ±20% on the final atomic proportions. Given the wetting of the metals on the substrates, NPs naturally form by atom aggregation upon thermal evaporation. Each run included at least two types of substrates: cleaved Si (001) samples, for NW growth, and carbon-coated gold TEM grids for the characterization of the NPs by TEM. Specific runs included in addition a Protochips™ SiC heating membrane for the in situ TEM experiments (Protochips, Inc., Morrisville, NC, USA).

The substrates undergoing the standard ex situ treatment were then loaded into a PECVD reactor operating at a radio frequency (RF) of 13.56 MHz for the fabrication of SiNWs. A two-step NW growth process was applied to the as-deposited Cu-In NPs after the vacuum reaches 5 × 10^−5^ mbar in the PECVD chamber. A schematic diagram of a SiNW growth experiment in a PECVD reactor is shown in Appendix A. In this reactor, the actual temperature is deduced from the nominal one through an abacus [41]; the error which is made depends on the substrate and is in the range of ±10 °C. However, we could check that such a range does not change the mechanism at play during SiNW growth. The process includes two steps:-Step I—hydrogen plasma pre-treatment at 200 °C. For this step, the H_2_ flow rate, the gas pressure *P*, the RF plasma power density and the duration time *t* were 100 sccm, 0.8 mbar, 56.7 mW/cm^2^, and 2 min, respectively. To evaluate the effect of plasma, we compared the Cu-In NPs after the plasma treatment with the NPs having experienced the same temperature annealing but without plasma. To follow the effect of plasma in real time, we made the same experiment in situ, in the TEM (with smaller catalysts deposited on the Protochips™ heating membranes);-Step II—SiNW growth at 420 °C. For this step, a second plasma treatment was carried out in the same run in the PECVD reactor by adding a SiH_4_ flow of 5 sccm to 100 sccm H_2_ and changing the process parameters to the following values: *P* of 1.42 mbar, RF power density of 17 mW/cm^2^ for *t* = 3 min. To understand the effect of the plasma and that of adding silane, we compare the Cu-In catalyst NPs at the head of the obtained SiNWs with (i) those NPs that have undergone exactly the same plasma pre-treatment at 200 °C as mentioned above and the 420 °C anneal treatment but the latter in vacuum, and (ii) the NPs that have undergone the 420 °C plasma treatment, but with only hydrogen and no SiH_4_ in the plasma (the other experimental parameters remain unchanged as in step I).

Table 1 lists the detailed parameters of six groups of experiments of plasma treatment and SiNW growth in this study. Groups 1 to 5 were carried out in a PECVD reactor named Plasfil, using a bimetallic catalyst with a nominal thickness of 1 nm In/1 nm Cu on carbon-coated gold grid substrates for the convenience of TEM characterization. Group 6 was an in situ TEM experiment using 0.6 nm In/0.2 nm Cu as a catalyst on a SiC TEM heating membrane. Additionally, groups 7 to 10 were also performed in Plasfil PECVD reactor, using the same experimental condition as in group 3, but on (001) c-Si substrates.

The structures of the Cu-In NPs before and after the plasma treatment were characterized using two TEM microscopes for high resolution imaging and selected area diffraction patterns (SADP): a Jeol 2010F (Jeol Ltd., Tokyo, Japan) and a Thermo Fisher Titan 80-300 named “Nan’eau” (Thermo Fisher Scientific Inc., Waltham, MA, USA). The acceleration voltages utilized were 200 kV for the Jeol 2010F TEM, and 300 kV for the Titan TEM. The diffraction patterns are interpreted with the help of the JEMS software, version 4.11531U2022b31 [45,46] and CIF files from the Inorganic Crystal Structure Database—ICSD. After growth, SiNWs are observed by scanning electron microscopy (SEM) using a Hitachi S-4800 (Hitachi High-Tech Corp., Tokyo, Japan). Some SiNWs are transferred from the c-Si substrates to carbon-coated gold TEM grids for Energy Dispersive X-ray spectroscopy (EDX) analysis in TEM Nan’eau.

The in situ TEM study of the Cu-In catalysts was carried out in an environmental TEM (ETEM) named “NanoMAX”: it is a modified Thermo Fisher Titan 80-300 equipped with an aberration corrector, which allows a high resolution of 0.08 nm in TEM mode; it is also equipped with a Gatan UltraScan 1000 camera (Gatan Inc., Pleasanton, CA, USA) which can record in situ videos at a rate of four 1 k × 1 k frames per second. Heating is performed using the SiC heating membranes of the Fusion sample holder from Protochips™. Each heating chip is delivered with a digital abacus giving the temperature as a function of the applied power. We measured an error of ±5 °C on the melting points of pure Sn and pure In large particles (theoretical points at 157 °C and 232 °C, respectively). In most of the chips, no temperature variations could be detected over the utilized part (~50 μm)^2^ of the heated area. The NanoMAX TEM is equipped with an Aura-Wave electron cyclotron resonance (ECR) plasma source from SAIREM (Décines-Charpieu, France), which is situated at about 0.4 m from the sample. We used a plasma power of 50 W. The recombination of ions and electrons takes place very fast, so that no ion gets to the sample [47,48]. However, the recombination of H atoms to make H_2_ molecules is much less efficient, so that the presence of hydrogen neutral atoms remains significant at the sample level [47,48]. In the in situ experiments, Steps I and II were replaced by, respectively, a Step I’ where the substrate was heated up to 250 °C and the H_2_ flow, injected into the plasma chamber connected to the TEM column, was set to 30 sccm, and a Step II’, where the substrate temperature was set to 370 °C and the SiH_4_ flow to 1.5 sccm; the latter gas was injected directly into the microscope (not through the plasma chamber). The pressure in the column is 3 × 10^−2^ mbar during growth.

## 3. Results and Discussion

### 3.1. Evolution of Bimetallic Cu-In Catalyst NPs throughout the Process

We observed the structure of Cu-In NPs in the as-deposited state and throughout the SiNW growth process. The standard In/Cu thickness we use is 1/1 nm (nominally 69.6 at.% Cu), unless otherwise mentioned. Given the uncertainties mentioned in the experimental section, the actual Cu percentage may vary, especially between deposits made with a several-month interval by different experimentalists, between ~40% and ~90%. The as-deposited NPs are mostly amorphous, with diameters ranging from 4 to 10 nm, as shown in Figure 1a.

#### 3.1.1. 200 °C H_2_ Plasma Treatment

We firstly performed the hydrogen plasma pre-treatment at 200 °C for 2 min in the PECVD reactor on the as-deposited NPs (see Table 1, Group 1). The bimetallic catalyst NPs adopt a core–shell structure: a crystalline core is covered by an amorphous shell (Figure 1b,c). The diameters of the catalyst NPs after such a hydrogen plasma treatment range from 4 to 24 nm (see Section 3.1.2, the comparison between the distributions after the 200 °C and 420°C treatments). A SADP on a larger area exhibits diffraction rings that are best fitted by the high-temperature phase Cu_4_In-β (see Appendix A, ICSD file #109480) [49], normally stable above 574–576 °C [50,51,52]. The high-resolution image of a NP in the [001] zone axis exhibits the typical square pattern of the cubic structure of this phase (Figure 1c,d). Let us note, however, that the β-phase thus defined differs from the α-phase (solid solution in fcc Cu) by only small atomic displacements—in a way quite similar to that found between the γ (fcc) and α (bcc) phases of iron—where ordering of In atoms would bring a new line at 0.3 nm. Annealing the as-deposited Cu-In NPs at 200 °C for 2 min (see Table 1, group 2) without plasma did not result in such a drastic reorganization nor crystallization: the NPs underwent some coalescence but kept a worm-like shape (see Appendix A). Thus, indium stayed amorphous upon deposit and plasma treatment, while copper crystallized in quasi-spherical shapes during annealing, incorporating some of the In, only if exposed to the plasma; In kept sticking to Cu in all cases.

The goal of the in situ experiment in NanoMAX (see Table 1, group 6) was to image the dewetting and coalescence of the NPs. The deposit used in this case concerns smaller amounts of In and Cu: (resp. 0.6 nm and 0.2 nm; nominally 42 at.% Cu), with the aim of forming smaller catalysts with a larger liquid part, better suited for obtaining the 2H-phase in the Si NWs made with them [19]. It was performed at 250 °C and 3 × 10^−2^ mbar H_2_. As-deposited NPs are hardly visible in the TEM (Figure 1d); switching on the plasma triggered their dewetting almost instantaneously, and drastically increases their TEM contrast (Figure 1e) (see also Appendix A). We associate this effect with a fast reduction of the In surface oxide, together with a change in the surface energy balance, due to coverage by adsorbed H atoms. Perhaps due to a lesser amount of matter, and to a surface quite different from that of amorphous carbon, we observe very little coalescence with such small particles.

#### 3.1.2. 420 °C Plasma Treatment and Si Nanowire Growth

After the hydrogen plasma pre-treatment at 200 °C, we carried out SiNW growth at 420 °C by exposing the Cu-In NPs to a hydrogen-silane plasma for 3 min and analyzed the structure of the Cu-In NPs after the initial stage of SiNW growth (see Table 1, group 3). SiNWs with Cu-In catalyst NPs on top are obtained after this process. Figure 2a,b shows a high-resolution TEM (HRTEM) image of a catalyst NP of 1 nm In/1 nm Cu after the synthesis of SiNW. The FFT image of the catalyst (Figure 2c) indicates spots corresponding to 0.78, 0.39, 0.26, and 0.20 nm interplanar distances, which represent a ~2% dilatation compared to interplanar spacings of (−110), (−220), (−330), and (−440) planes of Cu_7_In_3_ (𝛿) [53]. In addition, we note, close to the interface with the SiNW, the presence of shorter spacings, making moirés when superimposed with the former in Figure 2a,b, and generating additional spots in the FFT (blue arrows in Figure 2c). These spacings belong to the (002) (0.37 nm) and (003) (0.24 nm) planes of Cu_3_Si-η. We thus have a clue of the presence of Cu_3_Si, which is usually found in the catalysts that contain copper during the growth of SiNWs [18,19,54]. Moreover, no solid evidence of other intermetallic phases of Cu and In or their oxides have been found in the FFT patterns. We performed EDX analysis on a catalyst NP after SiNW growth to verify its composition (see Appendix A). The detected atomic percentage of Cu/(Cu+In) was about 72%, which is close to the nominal Cu at.% value (69.6%) in the 1 nm In/1 nm Cu deposit used, and is also close to the nominal Cu at.% value (70%) in the Cu_7_In_3_ (𝛿) phase.

In order to try and better understand the role of plasma, we have set a control group in step II in order to verify that the formation of the Cu_7_In_3_ (𝛿) phase only results from the hydrogen-silane plasma treatment rather than the elevated temperature (see Table 1, group 4). The TEM results, displayed in the Appendix A, show that the catalyst NPs that were annealed at 420 °C for the same time duration, but in vacuum, without being processed by plasma, kept the same crystalline structure of Cu_4_In as those obtained from step I. Weaker rings in Appendix A could be due to other Cu-In intermetallic phases, but the majority of the diffracted intensity undoubtedly comes from the Cu_4_In formed earlier.

However, as we know that catalysts undergo major changes during cooling of the specimen [18], we tried to observe the growth of SiNWs in situ using the NanoMAX microscope (see Table 1, group 6). We kept for this the 0.6 nm In/0.2 nm Cu deposit shown in Figure 1e,f. Observing growth in an exact zone axis was not possible, but a SiNW was oriented so that the Si (111) planes perpendicular to the growth direction were visible (Figure 3). The crystalline contrast in the catalyst demonstrates that at least a part of the NP is crystalline. In other frames (see Appendix A), roundish shapes and absence of contrast indicate that the catalyst may also have a liquid character. Therefore, as in the case of Cu-Sn [18,19], we could have a liquid-assisted vapor–solid–solid (LAVSS) growth mechanism. Quite strangely, the interplanar distance visible in the catalyst in Figure 3b varies continuously, from 0.38 nm close to the interface with the SiNW, to 0.27 nm at the top. It is thus difficult to conclude on the phase(s) present; let us note that the 0.38 value sits in between the 0.37 and 0.39 nm measured after ex situ growth (see above).

It is mentioned above that the Cu_7_In_3_ (𝛿) phase was obtained after SiNW growth with a hydrogen-silane plasma at 420 °C. In order to verify which gas plays a major role in this plasma-induced phase transformation phenomenon, we processed the as-deposited Cu-In NPs only in hydrogen plasma at 420 °C for 2 min in the PECVD reactor, applying NW growth conditions but without SiH_4_ (see Table 1, group 5), in such a way as to isolate the effects of the SiH_4_ on the catalyst microstructure (Figure 4). The Cu-In NPs obtained after a hydrogen plasma treatment at 420 °C have diameters mostly ranging from 4 to 20 nm. (Figure 4a,d). We have obtained the Cu_11_In_9_ structure [55] (see Appendix A, and ICSD file #238715) which is both different from the Cu_4_In found at 200 °C (Appendix A) and from the Cu_7_In_3_ found when silane is introduced in the plasma at 420 °C (Figure 2). Figure 4b shows a HRTEM image of a phase-separated Cu-In NP, and Figure 4c displays the FFT of its left-hand part (red-squared area in Figure 4b). The FFT pattern is that of the [001] zone axis of Cu_11_In_9_, with spots of the (200), (−110), and (110) atomic planes of Cu_11_In_9_ (respective interplanar distances of 0.52 nm and 0.40 nm). The crystalline phase on the right part of the NP shows atomic planes with an interplanar distance of 0.21 nm, which can belong to the (110) planes of Cu_4_In or the (111) planes of Cu. Thus, switching to the SiNW growth conditions, but without SiH_4_, promotes the development of the metallic Cu_11_In_9_ phase in the nanoparticles, with a possible switch of the remaining original phase from Cu_4_In to α-Cu. Quite interestingly, we thus observe a relative decrease in Cu content in the compound, compared to the SiH_4_ case (Cu_11_In_9_ vs. Cu_7_In_3_), while the missing copper stays in the NPs under the form of Cu:In-α or Cu_4_In-β, in the present case, and under the form of Cu_3_Si in the silane case (Figure 2).

#### 3.1.3. Catalyst Evolution: Summary

We can now summarize the structural evolution of Cu-In bimetallic catalyst NPs (1 nm In/1 nm Cu, nominally 69.6 at.% Cu) throughout the process (Figure 5). During thermal evaporation, Cu is firstly evaporated on the substrate and then In is evaporated on top of Cu. According to the Cu-In binary phase diagram [22,50,51,52], Cu_7_In_3_ (𝛿) is the stable phase at room temperature under the circumstance of 70 at.% Cu nominal content. However, the Cu_7_In_3_ (𝛿) phase does not form in the process of thermal evaporation. The deposit obtained upon thermal evaporation has an essentially amorphous structure covered by In native oxide, including only a few Cu crystals (Figure 1a and Figure 5). That structure remains so when the 200 °C anneal is not performed in the hydrogen plasma. The crystallization of Cu_4_In detected after plasma treatment (Figure 1b–d) is thus promoted by the plasma, not the temperature. In the core–shell structure appearing after this treatment, one can evaluate the atomic ratio in the particles from the volume ratio of the shell in Figure 1c. The latter represents about 59% of the total volume of the NP. Taking the atomic densities of Cu_4_In (ICSD file #109480) and of crystalline In (ICSD file #639814), one finds that the In atoms in the shell represent 43% of all the atoms. Adding the In atoms present in the core leads to 54% In, which value represents a maximum, as amorphous In is probably less dense than crystalline In. Overall, such an evaluation appears compatible with the nominal 30% in the deposit, as the latter is subject to the large errors mentioned earlier.

When the amorphous Cu-In NPs are processed with the two-step plasma conditions during SiNW growth in the PECVD reactor, we can summarize the different phase changes as follows (see Figure 5):

**Figure 5 nanomaterials-13-02061-f005:**
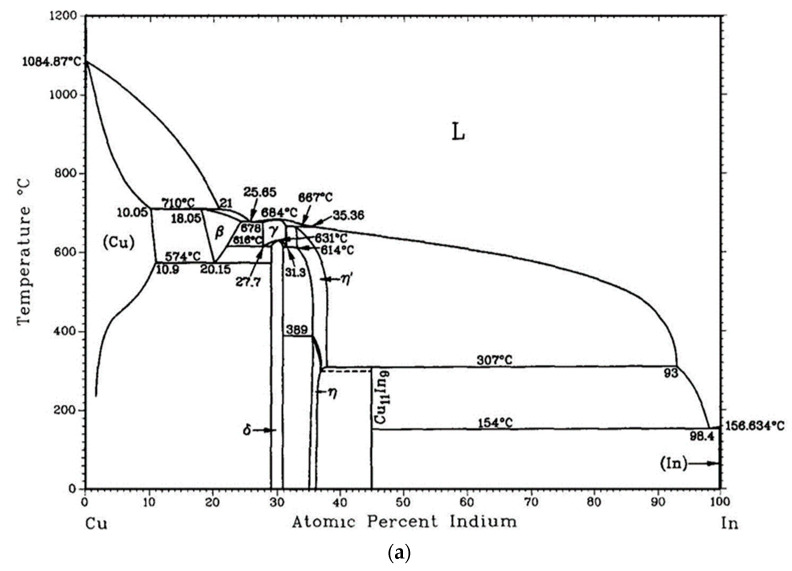
(**a**) Cu-In phase diagram [22], by courtesy of *Journal of Phase Equilibria*, copyright Springer Nature. (**b**) Summary of the evolution of the catalyst NPs after the different treatments. In is oxidized by air after deposition. Both In and the oxide are amorphous at this stage. The crystalline cores are very limited in extent, so that a part of the copper can also be amorphous. The 200 °C annealing in H_2_ plasma reduces the oxide and helps the cores to form larger crystals, in contrast to the same annealing without plasma. The SiNW growth conditions—420 °C (group 3), hydrogen-silane plasma—generate phases different from those obtained with the same annealing but without silane in the plasma, different themselves from those obtained with the same annealing in vacuum.

I. During the hydrogen plasma pre-treatment at 200 °C (group 1), Cu crystallizes under the effect of hydrogen plasma. Let us start by observing that, at this temperature, the In oxide which must be present at the NP surfaces is reduced by the H_2_ plasma. The H-plasma-induced reduction of In oxide is well known indeed in ITO (indium–tin oxide) [56]. Considering that 200 °C is higher than the melting point of pure indium (156.63 °C), the reduced In melts during this step and forms a liquid layer outside the Cu_4_In crystal core (it stays amorphous upon cooling, hence its structure in the present TEM observations). If we assume that the surface energy of Cu_4_In remains close to that of pure Cu, we may consider that the surface energy of the NP core stays above 1 Jm^−2^ [57], during the whole H_2_-plasma treatment at 200 °C, while that of In remains neatly lower, around 0.5 Jm^−2^ [57]. During the treatment, the liquid indium thus makes a continuous shell around the Cu-based cores, without leaving the core surfaces exposed to the plasma. However, the phase transition to crystalline Cu_4_In, which does not take place at 200 °C when the plasma is off, remains surprising. We tend to think that the H atoms provide two things: (i) they reduce the native In oxide and (ii) help break the bonds between disordered metal atoms; both of these facts lead to an enhanced atomic mobility that corresponds to a lowering of the potential barriers between the different phases. Moreover, the equilibrium structure in a NP depends on its surface energy, which, here, is determined by the In coverage. We would propose that the high-temperature Cu_4_Si phase (the β-phase in the phase diagram, Figure 5a) would be stabilized by this specific particle surface and the fact that In atoms can easily escape liquid In, thanks to the presence of atomic hydrogen, which helps them to enter the Cu structure.

II. By performing the hydrogen plasma treatment on the Cu-In NPs at 420 °C (group 5), the intermetallic phase Cu_11_In_9_ is obtained (Figure 5). Since Cu_11_In_9_ does not exist above 310 °C according to the phase diagram [22], we infer that it was formed during cooling. In the control experiment carried out at 420 °C in a vacuum (i.e., without plasma), only very little intermetallic compound was formed and Cu_4_In was essentially preserved. Thus, H atoms generated in the plasma diffuse through the liquid In and help destabilize the metastable Cu_4_In (let us recall that Cu_4_In is stable only above ~575 °C); In is then able to penetrate the new intermetallic compound, which, following the phase diagram [51], would be a mixture of In and Cu_7_In_3_ (𝛿). Cu_11_In_9_ precipitates when cooled down. Since the nominal Cu at.% is 69.6%, there are residual copper atoms and they exist in the form of Cu crystals as shown in Figure 4b. Thus, hydrogen plasma at 420 °C plays an important role in the synthesis of Cu_11_In_9_.

Why, now, is it the intermetallic phase Cu_7_In_3_ (𝛿)—and not Cu_11_In_9_—that remains after cooling, when SiH_4_ is introduced in the 420 °C step? In this case, the in situ TEM experiment shows that the SiNW growth follows, at least partially, a VSS mechanism (Figure 3). However, if the mechanism is similar to that observed with Cu-Sn catalysts [18,19], the solid phase would be essentially made of Cu_3_Si, which is detected after cooling (Figure 2) and is compatible with a part of the interplanar spacings measured in Figure 3. The presence of Cu_3_Si in the samples indicates that a significant amount of In is left aside during growth, maybe in the form of an In-rich Cu-In compound or, most probably, in the form of Cu-saturated liquid In. Given the fact that Cu solubility in liquid In falls from ~9% at 420 °C to ~0 at 20 °C [22], in such a scenario, a significant amount of copper would be released during cooling, which would provoke the change of the In-rich Cu-In compound into the Cu-rich Cu_7_In_3_ (𝛿) phase. Additionally, Cu_3_Si remains present indeed, as it is stable at room temperature. In contrast, when no Cu_3_Si is present, as in the plasma-treated sample without silane, the copper probably stays in its own phase at 420 °C, at equilibrium with an In-rich Cu-In compound. In this case, there would be no copper atom available to enrich the Cu-In compound during cooling; the latter would, in turn, precipitate in the form of the Cu_11_In_9_ phase, poorer in Cu.

In this study, plasma treatment has been proved to be an efficient method to induce crystallization and reaction for the synthesis of the intermetallic phases of copper and indium in the catalyst NPs. Compared with heating, the introduction of atomic hydrogen provides more energy to facilitate the transition of the Cu-In NPs from disorder to order. The H_2_-plasma treatment at only 200 °C delivers the Cu_4_In-β phase normally stable above 574 °C. In addition, in the case of diffusion soldering using pure Cu and pure In solders, it takes 16 days for the formation of Cu_11_In_9_ at 290 °C; and it takes 1.5 h to synthesize Cu_7_In_3_ at 430 °C [40]. The reaction rate of Cu_7_In_3_ in the present case is 30 times faster at a lower temperature (420 °C). By controlling the specific plasma conditions in the PECVD reactor, we managed to control the reactions of the catalyst elements, obtaining various reaction products with a much faster reaction rate. This opens a new way of synthesizing novel alloy catalyst NPs and controlling SiNW growth by using a simple, convenient, fast, and low-cost method combining thermal evaporation and PECVD, which can realize both phase transformation of catalyst NPs and NW growth in the same run of PECVD process.

### 3.2. Influence of Catalyst Composition on the Distribution and Structure of SiNWs

In the following section, we focus on the characterization of the morphology, distribution and crystalline structure of the SiNWs fabricated with Cu-In bimetallic catalyst NPs, as a function of catalyst composition, using SEM and TEM. When the SiNWs are synthesized with the 1 nm In/1 nm Cu composition studied above (group 7 to 10), the NW diameter shows a broad distribution: from about 10 nm to 38 nm (Figure 6a,e). By decreasing the catalyst’s nominal thickness to 0.1 nm In/0.1 nm Cu (Figure 6b), the NW diameters have a narrow distribution and are mostly below 12 nm (Figure 6e). The longest NWs have lengths around 300 nm. Meanwhile, in the control group where pure In catalyst was used with a nominal thickness of 0.1 nm, the SiNWs obtained are shorter, thicker, and much less dense (Figure 6c,e). Previous experience indicates that SiNWs do not grow with pure Cu catalyst under the current conditions [19]. Using Cu-In bimetallic catalysts allows one to adjust the SiNW density, diameter distribution, and growth rate in a range which cannot be realized by using pure metal. Further decreasing the catalyst nominal thickness to 0.05 nm In/0.05 nm only reduces the NW density, while the average diameter (10.5 nm) remains similar (Figure 6d).

Regarding the structure of the SiNWs themselves, we searched for the metastable hexagonal 2H polytype of Si [21], particularly in the sets where the NWs were the narrowest [19]. For this, we observed more than 50 SiNWs, grown with 0.1 nm In/0.1 nm Cu, by TEM. Among these, 11 SiNWs were in the [110]_C_/[1−210]_H_ zone axis, which is the only zone axis allowing to distinguish the diamond hexagonal Si phase from the standard diamond cubic Si [44]. Ten of them had cubic structures, while one of them contained hexagonal phase. The SiNW with hexagonal structure (Figure 7a) has a crystalline core diameter around 5 nm, and a total diameter of 7.5 nm, which includes an amorphous shell. We found there is a catalyst separation phenomenon in this SiNW: in addition to the catalyst NP connecting to the SiNW crystalline core, there is still an amorphous part above it, on top of which lies another nanoparticle. Catalyst separation exists not only in the hexagonal NW in Figure 7, but also in the cubic SiNWs synthesized with the mixed catalyst of 0.1 nm In/0.1 nm Cu. Based on our experience on thin SiNWs catalyzed by Cu-Sn co-catalysts [58], we assume that a similar catalyst separation phenomenon has happened after SiNW growth, in the ambient air due to the use of Cu as a catalyst component. An HRTEM image with higher magnification of the selected area in Figure 7a is shown in Figure 7b. The crystalline structure consists of two parts: above the yellow dashed line (which is parallel to the SiNW growth direction) there is a cubic twin with the twin plane indicated by the white dashed line. Below the yellow dashed line, there is a hexagonal phase, which has been confirmed by the FFT pattern shown in Figure 7c, with $d_{\left(0002\right)}=0.314\;\mathrm{nm}$ and $d_{\left(10-10\right)}=0.336\;\mathrm{nm}$. The co-existence of cubic twinning and hexagonal Si does extend to the top of the crystalline core. This configuration, of the 2H structure in a SiNW with a [211]_C_/[10−10]_H_/growth axis, was quite rare in our previous observations of that phase in Cu-Sn catalyzed SiNWs [19,44], where the dominant configuration was with a [0001] growth axis, and where the present geometry only occurred after a right-angle change in growth direction [44]. Thus, the mechanism of formation of that metastable phase would be original in the present case, perhaps involving the atomic structure of the Cu_7_In_3_/Si interface at the start of the VSS growth: indeed, the (002) interplanar spacing of this triclinic structure matches quite remarkably (0.13% difference between the equilibrium phases) the spacing of (10−10) planes in 2H Si. With a proper catalyst orientation, the growth of the SiNW in Figure 7 would have proceeded, at the catalyst–NW interface, with the replacement of (002) planes in the catalyst by the (10−10) planes of the silicon hexagonal structure. After the formation of Cu_3_Si in the catalyst, the growth, taking place parallel to the dense (0002) planes of 2H-Si, would have pursued the 2H-stacking of these planes.

In addition, we also characterized using TEM the crystalline structures of thicker SiNWs synthesized with a mixed catalyst of 1 nm In/1 nm Cu (Figure 6a,d). All of the observed SiNWs had diameters larger than 15 nm and all of them showed cubic phase.

## 4. Conclusions

We have studied the structural evolution of thermally evaporated Cu-In bimetallic NPs throughout the SiNW growth process, with in situ as well as ex situ TEM observations. The ex situ process includes a hydrogen plasma pre-treatment in the PECVD reactor at a temperature of 200 °C and the growth process itself, in a H_2_-SiH_4_ plasma, in the same reactor at 420 °C. We show that the 200 °C pre-treatment induces a partial crystallization of the essentially amorphous as-deposited NPs, leading to a structure including a crystalline core (Cu_4_In)-β and a liquid shell (In); we also show that applying the hydrogen plasma treatment at SiNW growth temperature (420 °C), but without SiH_4_, induces the formation of the intermetallic phase Cu_11_In_9_ in the NPs. Quite interestingly, adding silane to the plasma changes that phase found after growth to Cu_7_In_3_ (𝛿). The in situ experiments show that the SiNWs synthesized with Cu-In bimetallic catalyst NPs with the hydrogen-silane plasma follow an essentially “VSS” PECVD process, but does not rule out the presence of a liquid part in the catalyst. By adjusting the composition of the bimetallic Cu-In catalyst NPs, we have obtained dense SiNW arrays with a smallest average diameter of around 10 nm and a narrow distribution that we could never realize with pure metal catalysts. Moreover, we were able to fabricate a SiNW (crystalline diameter of 5 nm) that includes a large hexagonal domain (which is predicted to have a direct bandgap). We believe that further studies on compound catalysts in general, and especially liquid–solid ones, could lead to a fine control not only of SiNW shape and density, but ultimately of their crystalline phase, opening up new prospects for crystal-phase heterostructures at the nanoscale.

## Figures and Tables

**Figure 1 nanomaterials-13-02061-f001:**
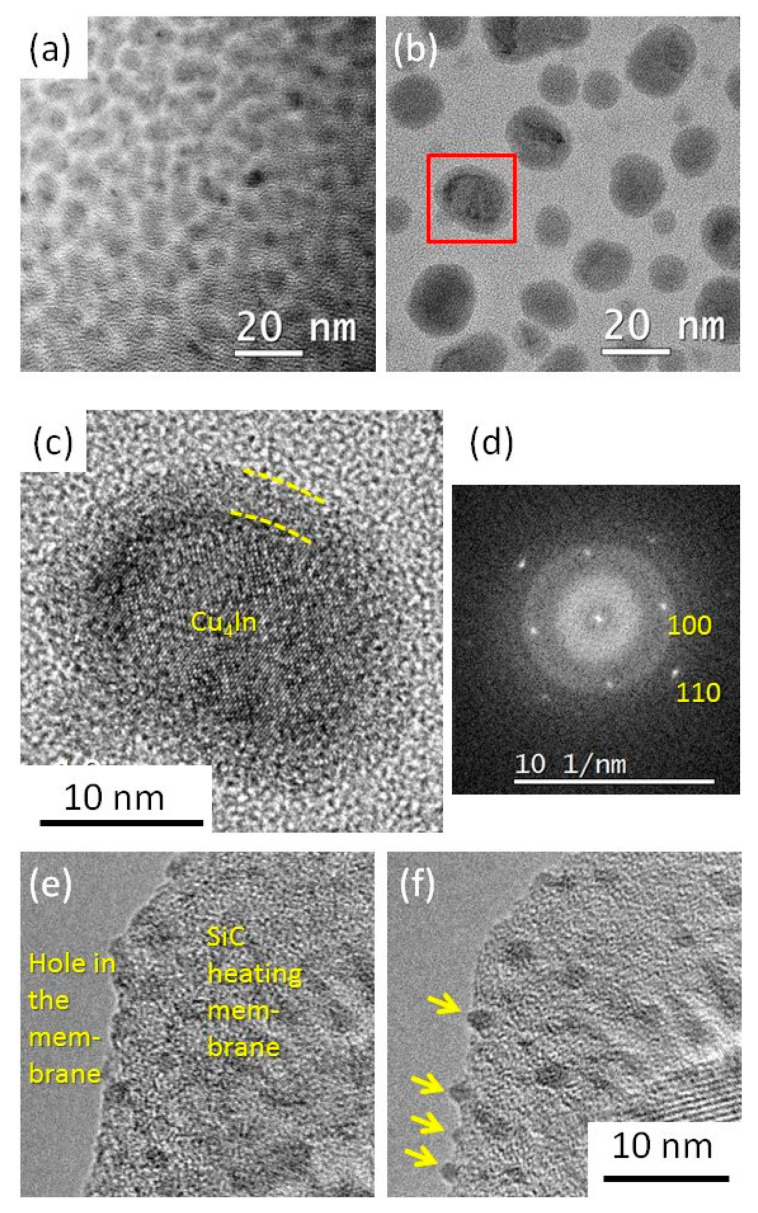
(**a**–**c**) TEM images of the bimetallic catalyst NPs obtained with a nominal thickness of 1 nm In/1 nm Cu: (**a**) as-deposited catalyst NPs; (**b**) NPs after a hydrogen plasma pre-treatment at 200 °C for 2 min (group 1); and (**c**) enlargement of the red-squared area in (**b**); (**d**) the power spectrum of the fast Fourier transform (FFT) of (**c**), exhibiting the pattern of the [001] zone axis of Cu_4_In. (**e**,**f**) Catalyst NPs obtained on a SiC TEM heating membrane with a nominal thickness of 0.6 nm In/0.2 nm Cu before and after exposing them at 250 °C to a hydrogen plasma in situ in the TEM (group 6); the yellow arrows in (**f**) show the plasma-dewetted Cu-In NPs.

**Figure 2 nanomaterials-13-02061-f002:**
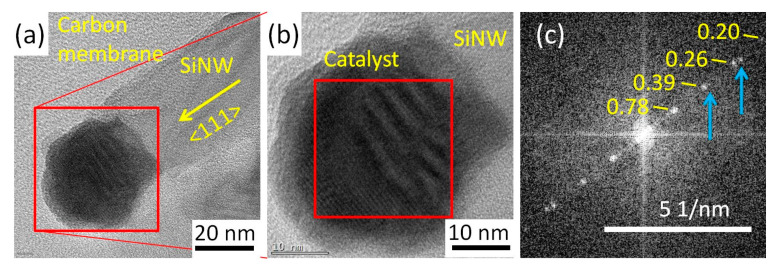
(**a**) HRTEM image of a Cu-In catalyst NP at a SiNW tip, obtained after SiNW growth at 420 °C for 3 min (group 3); (**b**) enlargement of the catalyst area in (**a**); (**c**) power spectrum of the FFT of the red-squared area in (**b**): the numbers indicate the lattice spacings (in nm) corresponding to the main spots, belonging to Cu_7_In_3_-δ; the blue arrows indicate secondary spots generated by the presence of Cu_3_Si.

**Figure 3 nanomaterials-13-02061-f003:**
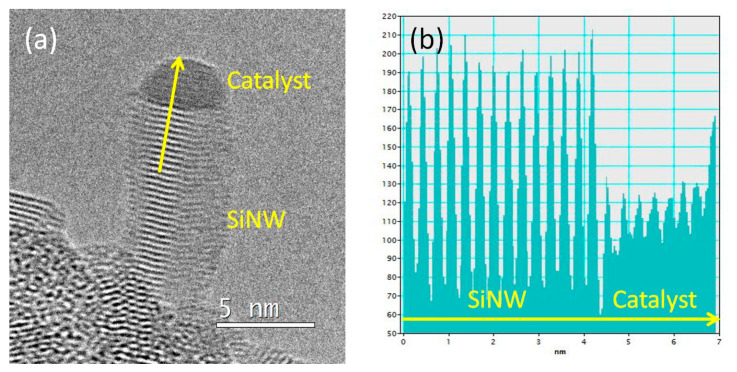
(**a**) SiNW growing in situ (group 6) in the NanoMAX TEM (frame from Appendix A), with a catalyst from the In-0.6 nm/Cu-0.2 nm deposit shown in Figure 1e; (**b**), intensity profile along the yellow arrow in (**a**) exhibiting the presence of reticular contrast in the catalyst, demonstrating that at least a part of it is crystalline.

**Figure 4 nanomaterials-13-02061-f004:**
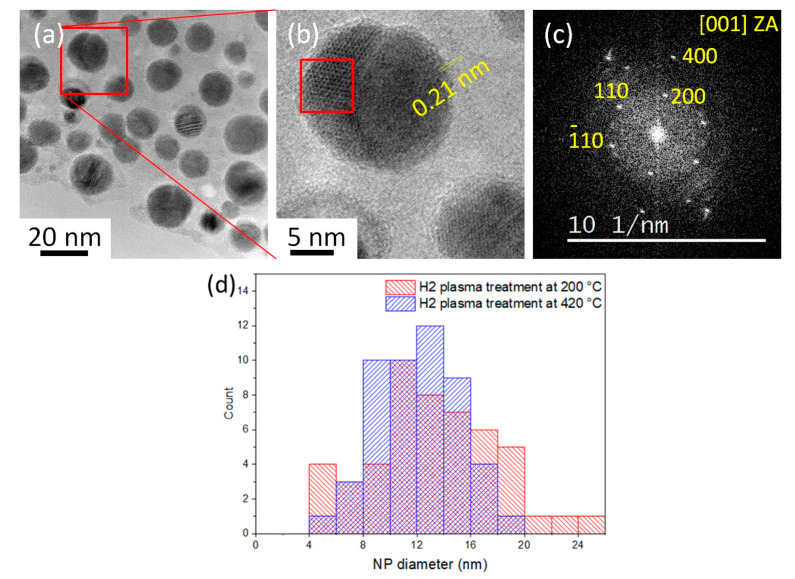
TEM of Cu-In NPs obtained after H_2_-plasma treatment at 420 °C for 2 min (group 5). (**a**) General view; (**b**) HRTEM image of the red square in (**a**), detailing a Cu-In NP with two separated phases; (**c**) FFT image of the red-squared area in (**b**), which we index in terms of the Cu_11_In_9_ [001] zone axis; and (**d**) histogram of diameter distributions of Cu-In catalyst NPs after hydrogen plasma treatment at 200 °C (group 1) and at 420 °C (group 5).

**Figure 6 nanomaterials-13-02061-f006:**
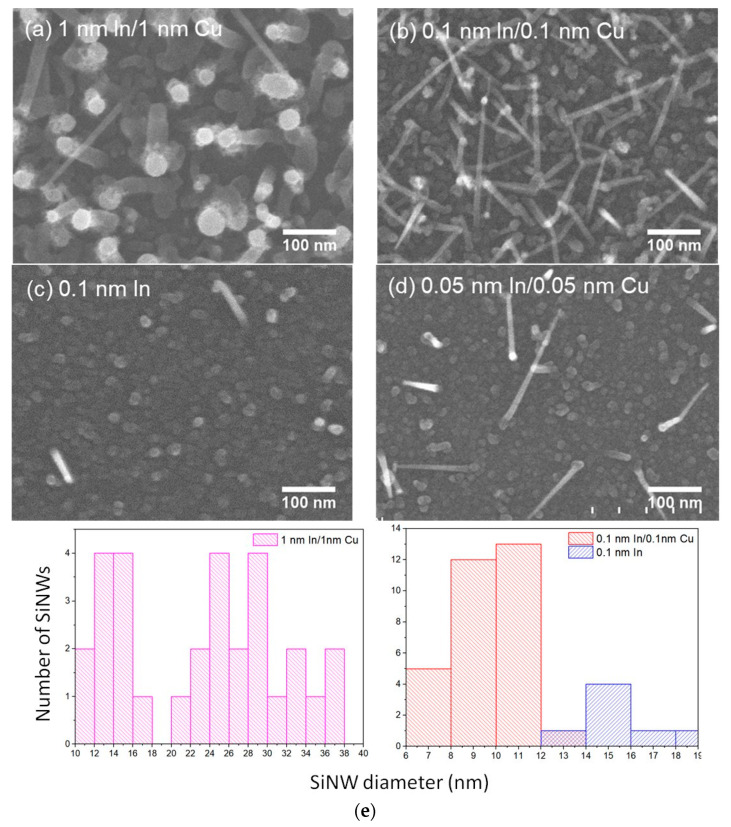
SEM images of SiNWs (groups 7 to 10 in Table 1) grown with mixed catalysts of (**a**) 1 nm In/1 nm Cu, (**b**) 0.1 nm In/0.1 nm Cu, (**c**) 0.1 nm In (**d**) 0.05 nm In/0.05 nm Cu on (100) Si substrates and (**e**) histograms of distributions of SiNW diameters.

**Figure 7 nanomaterials-13-02061-f007:**
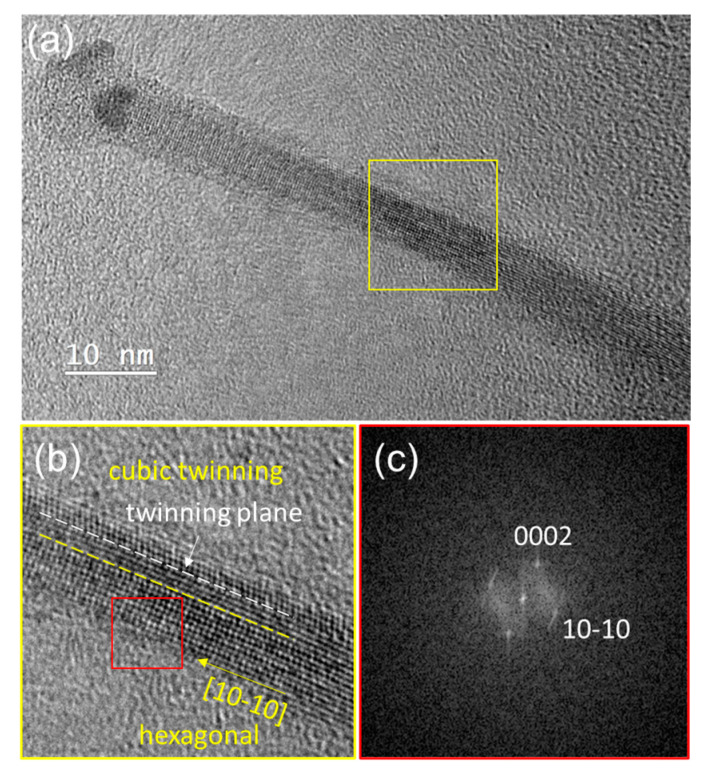
(**a**) HRTEM image of a SiNW synthesized with 0.1 nm In/0.1 nm Cu co-catalyst (group 8), transferred on a carbon-coated gold TEM grid from (100) Si substrate; (**b**) enlarged view of the yellow-squared area in (**a**), showing the co-existence of cubic twinning and hexagonal Si along the NW growth direction; (**c**) FFT image of the red-squared area in (**b**) showing the pattern of the [1−210] zone axis of hexagonal Si.

**Table 1 nanomaterials-13-02061-t001:** Detailed PECVD parameters of the plasma treatment and the SiNW growth.

			Step I: 200 °C				Step II: 420 °C			
Group	Catalyst	Substrate	*T* (°C)	H_2_ Flow Rate (sccm)	*P* (mbar)	Power Density (mW/cm^2^)	*T* (°C)	H_2_ + SiH_4_ Flow Rate (sccm)	*P* (mbar)	Power Density (mW/cm^2^)
1	1 nm In/1 nm Cu	carbon-coated gold TEM grid	200	100	0.8	56.7				
2			200	100	0.8	0				
3			200	100	0.8	56.7	420	100 + 5	1.42	17
4			200	100	0.8	56.7	420	0	1.42	0
5							420	100	0.8	56.7
6	0.6 nm In/0.2 nm Cu	SiC membrane	250	30		Power of 50 W	370	30 + 1.5	3 × 10^−2^	Power of 50 W
7	1 nm In/1 nm Cu	Si (001)	Growth parameters: same as group 3							
8	0.1 nm In/0.1 nm Cu									
9	0.1 nm In									
10	0.05 nm In/0.05 nm Cu									

## Data Availability

All the data are available upon request to J.-L.M.

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
