# Peer review of "Evolution of Cu-In Catalyst Nanoparticles under Hydrogen Plasma Treatment and Silicon Nanowire Growth Conditions"

_nanomaterials, 2023, doi:10.3390/nano13142061_

Round 1
Reviewer 1 Report
This is an excellent paper that deals with evolution of compound Cu-In catalyst nanoparticles under hydrogen plasma treatment and silicon NW (NW) growth conditions. The use of compound catalyst may generally influence on the growth behavior, morphology and crystal structure of VLS nanowires. However, a detailed analysis of this influence is known mainly for Au-Ag and Au-Ni catalysts, so the thorough study of the Cu-In nanoparticles catalyzing the VLS growth of Si NWs is new and important. The paper reveals some interesting effects, including a partial crystallization of the nanoparticles under hydrogen plasma treatment at 200 C, and the influence of a H2-SiH4 plasma treatment on the Si NW growth in a PECVD reactor. The conclusions of the paper are nicely supported by the TEM data and analysis. The paper is well-written and the citation list is relevant. I have found no weak points of the paper and can recommend its publication as is.
No problem with the English.
Reviewer 2 Report
The Article “Evolution of Cu-In catalyst nanoparticles under hydrogen plasma treatment and silicon nanowire growth conditions” is devoted to the study of the silicon nanowire growth with a novel Cu-In bimetallic catalyst using PECVD method. In this work, a precision study was carried out using TEM HR. It is shown that the synthesis of SiNW with Cu-In bimetallic catalyst NPs follows an essentially vapor-solid-solid process.
The article may be published in its present form. As a note, it is necessary to present a flowchart of the experiment due to the large number of different comparative experiments. It is also obvious from the presented TEM photographs that the resulting nanoparticles have different sizes, so it is interesting to see the size distribution diagram.
Authors sometimes use somewhat specific terminology, such as nano-solder.
Reviewer 3 Report
The paper reports on the use of Cu-In alloy catalysts to grow SiNWs. The potential co-existance of liquid and solid phases. The study is mainly based on TEM analysis. The paper itself is overall interesting but some aspects must be still discussed prior acceptance/publication:
- The according phase diagram should be shown and the discrepancy between thermodynamic equilibrium phases and the shown synthesis must be discussed, which might explain some of the experimental observations better. Is also supports the readability.
- hexaogonal Si is proclaimed and shown using TEM, nevertheless this part must be strengthened in the discussion based on the used catalyst to demonstrated the rationality of the synthesis strategy rather than just reporting experimental results and making claims.
- the ability to control the nanowire morphology, e.g. diameter must be supported by a statistical analysis and data plotting to demonstrate and clearly shown the kind of the control. Now there are only few TEM images
- The introduction part appears based on the emphasized motivation and references partly outdated. The number of references concerning alloy catalysts in VLS/VSS appears incomplete, e.g. look to Nano Lett. 2019, 19, 11, 7895–7900, https://doi.org/10.1021/acs.nanolett.9b02950. and also the Refs 13-17 listed there.
Reviewer 4 Report
I have read with attention your manuscript.
I found the phenomenology of the phase transformations of he bimetallic nanoparticles fairly well described, with very good TEM images to analyse the process.
It's unclear to me how important it is for the Si NW technology: but it's well done, with good starting data, and well analysed and described, so I think it is very worth publishing.
I have few general (major) comments that I would like the authors to consider, accompanied by a number of (minor) comments to specific points inthe MS, and a single typo. I strongly believe that the MS could be substantially improved with relatively little effort, and I hope the authors consider my comments to revise it. However, it's suitable for publication once the general A and C issues below are taken care of.
A. Process temperature: indicated as absolute values with no uncertainty. Some discussion or evaluation of the precision of the T determination would be useful. Particularly striking is the 416° C reported value for step II: if it is 416+/- 0.5°C as one would infer, I think you have to comment on such a very precise determination of T. For step I, 200°C could be +/-50 or +/5: can you explicitly state it? As for the in situ TEM, you mention the commercial name of the chip used, but the manufacturer website does not specify the accuracy, precision and uniformity of the temperature. Please add some detail.
B. Several substrates, with different deposited thicknesses are described in the MS, but in the discussion and figures it is seldom specified what substrate/deposition is being treated/described. Perhaps you could add a table in the methods, giving a short name to each substrate/thickness, to easily but clearly refer to them in the MS.
C. Throughout the MS you state that an effect appearing with the plasma ON and not appearing with the plasma OFF is a strong indication of a direct effect of plasma on the nanoparticles. Can you explicitly rule out the following:
1. An effect of the plasma on the actual catalyst or surface temperature, either lowering it by means of thermal conduction by the gas (P’s are fairly high), or increasing it by plasma energy transfer or radiation heating if the plasma irradiates a lot. Can you rule out these mechanisms by calculating their contribution and showing that it’s negligeable? Have you tried a lower or higher T thermal treatment? (see the first comment on accuracy of your T determination)
2. An effect of the plasma on the substrate surface, changing its energy and wettability. Have you tried plasma treating the substrates and then depositing the metal?
D. Specific issues:
Line 48-50: the fact that Cu is a p-dopant for Si is -imho- a reason NOT to use it as a catalyst, since you risk to mix doping distribution and density to morphology and structure. I think you should mention it as a negative side effect, and comment on the actual incorporation (possibly very little or none) focusing on the plus: the effect on crystal structure.
Line 81: Protochips SiC Heating membrane: no details are available on the manufactirer’s website. Please describe what they are.
Lines 88-90: annealing temperatures, with or without plasma. You state fairly short times, with no mention of transients in temperature or ramps. Can you please comment on how fast you ramp up the T and in what conditions, also going to step II? One could guess the the H2 plasma is on all the time, but perhaps you switch it off during the T ramps. Please specify.
Line 95: why is step II process pressure so much higher than step I? 100 SCCM of H2 plasma (~200 SCCM of H ions) gat a 0.8 mbar pressure, and adding a meagre 5 SCCM of SiH4 almost doubles it? Do you have a butterfly valve allowing you to adjust the process pressure independently from the flows? Please clarify.
Line 96. Can you comment on the much lower plasma power density of step II with respect to step I?
Line 109: are “gold TEM grids” the same “carbon coated” grids mentioned at line ~80? Please be consistent, or name them simply “grids” here.
Line 117-120: H2 plasma in the TEM setup. It is unclear to me how the H ions plasma becomes a neutral monoatomic H gas at the sample surface. Ref 40 doesn’t mention H2 plasma, analyses distances only up to 0.14 m from the source (against the stated 0.4 m of the manuscript) and only treats the (ionised) plasma density, with no mention of neutralisation of the ions. Please explain better what you expose the sample to and why and/or provide the appropriate reference
Line 120-125: Step I’ and II’ seem very different from steps I and II both in T and in plasma density or even gas introduced (monoatomic neutral H (see previous comment) gas in I’ against a H2 plasma in I, SiH4 gas in II’ against SiH4 plasma in II) Some explanation on why it supposed to be comparable is necessary. In particular growth of silicon at low temperature is definitely happening if you assist the SiH4 decomposition by plasma, but without plasma I’m not sure it’s normally working. Can you provide some references of low-temprature non-plasma assisted CVD of Si with Silane? Perhaps there’s an important role of the liquid part of the catalyst? Is this new (emphasize the finding!) or not new (provide references!)?
Lines 129-131: in a standard evaporator with a crystal balance having an approximately unitary tooling factor in general you have some Angstorms of uncertainty. Please comment on the 1nm/1nm thickness (and on the 0.6/0.2 nm thickensses later) and the three-significant-digits ”69.6 at% Cu” stated: it seems to imply that your thickness uncertainty is of the order of 0.001 nm or better, which I cannot believe. Please indicate your thickness uncertainty, and onsider rephrasing i.e. “nominally 69.6%” across the whole manuscript, or ~70%. Actually you discuss this at lines 150-154, mixing up thing a bit. Probably discussing the thiocknesses deposited and their uncertainties and the atomic fractions (with thei uncertainties) would be better done in the methods section. Same at line 169.
Lines 154-157: you draw conclusions that better go with the interpretation in section 3.1.3.
Fig 5: I understood the the “416°C” H2 plasma was following the “200°C” H2 plasma, while in the figure it appears that you go straight to the higher T plasma. Maybe a better description of each sample treatment, (including what happens in the temperature ramps, already mentioned) would be beneficial. It can be boring, but you can include it in the SI, perhaps.
Fig 6 is very very little informative: it’s hard to guess what is what in the SEM images, and one has to believe you when in the text you make statement about diameters and densities. Perhaps you can move the figure in the SI, leaving only your analysis in the MS. Perhaps in the SI you have some room to explain how you extract quantitative information form those blurred SEM images…
E. Typos:
Line 60: temperature; exploring -> temperature, exploring
Round 2
Reviewer 3 Report
The authors have revised the manuscript in a satisfactory manner. There are no further questions from my side.